# Prospecting Local Treatments Used in Conjunction with Antivenom Administration Following Envenomation Caused by Animals: A Systematic Review

**DOI:** 10.3390/toxins15050313

**Published:** 2023-04-28

**Authors:** Érica S. Carvalho, Isadora Oliveira, Thaís P. Nascimento, Alexandre Vilhena da Silva Neto, Brenda A. S. Leal, Felipe Q. Araújo, Bruno F. V. Julião, Andrea R. N. Souza, Andreza W. Abrahim, Bruna B. O. Macedo, Jéssica T. S. de Oliveira, Fan Hui Wen, Manuela B. Pucca, Wuelton M. Monteiro, Jacqueline A. G. Sachett

**Affiliations:** 1School of Health Sciences, Amazonas State University, Manaus 69050030, Amazonas, Brazil; 2Department of Teaching and Research, Dr. Heitor Vieira Dourado Tropical Medicine Foundation, Manaus 69040000, Amazonas, Brazil; 3Department of BioMolecular Sciences, School of Pharmaceutical Sciences of Ribeirão Preto, University of São Paulo, Ribeirão Preto 14040903, São Paulo, Brazil; 4Department of Teaching and Research, Alfredo da Matta Foundation, Manaus 69065130, Amazonas, Brazil; 5Butantan Institute, São Paulo 05501000, São Paulo, Brazil; 6Medical School, Federal University of Roraima, Boa Vista 69310000, Roraima, Brazil; 7Health Sciences Postgraduate Program, Federal University of Roraima, Boa Vista 69310000, Roraima, Brazil

**Keywords:** envenoming, venomous animals, local effects, local therapies

## Abstract

Envenomation caused by venomous animals may trigger significant local complications such as pain, edema, localized hemorrhage, and tissue necrosis, in addition to complications such as dermonecrosis, myonecrosis, and even amputations. This systematic review aims to evaluate scientific evidence on therapies used to target local effects caused by envenomation. The PubMed, MEDLINE, and LILACS databases were used to perform a literature search on the topic. The review was based on studies that cited procedures performed on local injuries following envenomation with the aim of being an adjuvant therapeutic strategy. The literature regarding local treatments used following envenomation reports the use of several alternative methods and/or therapies. The venomous animals found in the search were snakes (82.05%), insects (2.56%), spiders (2.56%), scorpions (2.56%), and others (jellyfish, centipede, sea urchin—10.26%). In regard to the treatments, the use of tourniquets, corticosteroids, antihistamines, and cryotherapy is questionable, as well as the use of plants and oils. Low-intensity lasers stand out as a possible therapeutic tool for these injuries. Local complications can progress to serious conditions and may result in physical disabilities and sequelae. This study compiled information on adjuvant therapeutic measures and underscores the importance of more robust scientific evidence for recommendations that act on local effects together with the antivenom.

## 1. Introduction

Venomous animals (e.g., snakes, insects, and arachnids) can inoculate venom in humans due to the specific mechanisms they employ to penetrate tissues. Indeed, bites or stings by these animals may promote local and systemic damage to the victim, resulting in important impacts in public health [1]. The severity of the envenomation depends on the amount of venom inoculated, the location of the bite, the victim’s systemic condition, and the length of time between the envenomation occurring and the administration of antivenom. The classification of the envenomation can range from mild local clinical manifestations to more severe systemic complications [2]. Thus, the severity of the envenomation will be responsible for the clinical outcomes, including induced physical disabilities and deaths [2,3,4].

The treatment for envenomation consists of the neutralization of the venom via antivenom administration at the recommended dosage, which is determined according to the severity of the case [3]. Nevertheless, antivenom is more efficient at controlling systemic signs than in neutralizing local induced effects such as edema, localized hemorrhage, and tissue necrosis [2,5]. Actually, for mild envenomation with only local manifestations, besides the antivenom therapy, analgesia and local cold compresses are recommended for pain relief [6,7].

Local damage can progress to dermonecrosis and myonecrosis, leading to tissue loss or even amputations [8]. Thus, in instances of envenomation that present local damage, it is important to use additional combined therapies aimed at restoring tissue homeostasis [5]. Initially, careful cleaning of the wound is fundamental to prevent secondary infection [8]. Due to the wide variety of microorganisms present on the victim’s skin and in the oral cavity of the venomous animal, it may also be necessary to combine antibiotic and anti-tetanus therapies [8,9,10,11].

Knowing the greater predominance of envenomation in rural areas, which are associated with poverty and work activities in the countryside, the use of alternative and traditional treatments to minimize the effects of venoms is frequent due to local culture [5]. Among the therapies, the use of medicinal plants, locally or even orally, stands out as an ancient practice in human history [12]. Moreover, in vitro and in vivo studies have shown that some plants have properties that can inhibit the activities of venom and local damage [13,14], while others have no effect [15], although further studies are still necessary. In addition, the use of non-invasive procedures (e.g., laser therapy) to treat local damage has shown to be very promising due to their anti-inflammatory, analgesic, and tissue-regeneration effects; and undoubtedly the application of laser therapy has resulted in positive results, principally in the reduction of the formation of edema, myonecrosis and leukocyte flow [16,17].

Therefore, identification of best practices for local adjuvant treatments could help to mitigate local damage caused by envenomation. This systematic review aimed to evaluate the scientific evidence related to local treatments that, in conjunction with antivenom, may reduce the local effects triggered by envenomation.

## 2. Results

An overview of the 39 included studies can be found in Table 1 and Table 2, which contain the titles, year of publication of the studies, the type of study, the country where they were carried out, and the main aim of each work. The classification of the studies was carried out according to the type of venomous animal: (i) snakes: 82.05%; (ii) insects: 2.56%; (iii) spiders: 2.56%; (iv) scorpions: 2.56%; and (v) others (jellyfish, centipedes, sea urchins): 10.26% (Figure 1A).

The aim of each analyzed study was to present a gamut of local treatments to be used in the lesions caused by venomous animals. The analyses identified conventional treatments (such as analgesia, compresses, and tourniquets) and topical/oral medications, including plants and other promising adjuvant treatments (Table 3). The systematic review also evaluated the therapies used in each study and their outcomes (Table 4).

## 3. Discussion

Local tissue injury is one of the main forms of damage caused by envenomation. However, the severity of the local effects depends on the venom type, the amount of venom injected, and the victim’s prior health problems. Tissue damage may lead to severe consequences, such as vascular degeneration and ischemia, which can culminate in limb amputation [57].

The local damage following envenomation is caused by the venom’s components (i.e., toxins) and these can lead to inflammatory signs, hemorrhage, and necrosis [17,58]. Indeed, there is a need to look into alternatives for local therapy, since little research has been carried out and applied in humans. On the other hand, there are in vitro and in vivo studies in the literature that support the development of additional research to discover effective local therapy alternatives. This review addresses several aspects of local treatment, as well as the use of questionable first-aid methods following envenomation.

Currently, the use of tourniquets is not recommended due to the possibility of developing gangrene, and neither is the prolonged use of cryotherapy [56]. Thus, the most important and recognized action is the prompt transport of the victim to medical care [3]. For snakebites, the time elapsed between the bite and hospital admission is the most crucial factor in the patient’s clinical outcome [2]. Medical treatment (i.e., antivenom), when delayed for more than 6 h, is associated with a greater risk of permanent sequelae and death [59]. In contrast, the tourniquets used in patients bitten by *Crotalus durissus* does not lead to any negative consequences; however, this does not apply to those bitten by *Bothrops* snakes and the use of this technique is contraindicated for any envenomation because it is not effective in preventing local complications [47]. Indeed, since 1988, the use of tourniquets after spider and snake envenomation is contraindicated, since tourniquets were observed to be responsible for a high proportion of tissue loss and permanent disability [8].

Other local treatments used to counter local venom-induced damage include cryotherapy and ice compresses for envenomation by jellyfish [50], centipedes [52], sea urchins [53] and scorpions [49], but these have not yet shown advantages for snakebite envenomation. In contrast, since the cold increases the vasoconstriction, some authors defend the use of warm compresses, since warmth can be beneficial for other kinds of envenomation such as those caused by *Physalia physalise*, (Portuguese man o’ war) [8,56,60]. However, it is noteworthy that in envenomation by scorpions, bees, spiders, ants, centipedes, and wasps, a cold compress may relieve some symptoms such as pain [8,49,60,61]. Indeed, it has been proven that ice packs, corticosteroids, antihistamines, oral analgesics, and local anesthetics were efficient in reducing local inflammatory symptoms and pain in cases of wasp, bee, and ant envenomation [61,62,63,64].

Regarding bee envenomation, antihistamines are not recommended because they cause drowsiness; thus, the use of non-steroidal anti-inflammatory drugs and cold compresses is recommended. Oral corticosteroids are indicated only when there are severe systemic manifestations; there is no evidence regarding their topical use, and injectable epinephrine has no benefits [61]. In another study, the authors recommend the use of epinephrine, oral and intravenous corticoid, and the application of cold compresses after bee stings [63].

For other envenomation by Hymenoptera, such as fire ants (*Solenopsis invicta*), the treatment of mild and severe envenomation (erythema, edema and pain) generally consists of conservative therapy, with indications of cold compresses, antihistamines, topical corticoids, topical application of lidocaine and, for symptomatic treatments, warm baths are indicated [63].

Regarding spiders, cleaning of the wound, raising the affected limb, immobilization, and cold compresses and analgesics are suggested as being effective [65] and, when there is no injury aggravation, the elevation and immobilization of the affected limb is recommended. In addition, for bites by the yellow spider, cold compresses and immobilization of the affected limb are indicated, as well as medication for the treatment of itching and analgesia (except acetylsalicylic acid) [65]. Other studies indicate that corticosteroids and intravenous antihistamines are not recommended because they are not effective against dermonecrosis, and the use of the antibiotic dapsone remains uncertain [60], which is different in cases of envenoming by the brown recluse spider (*Loxosceles inmates*), which vary from itching to death. In its most severe form, the bite of this spider causes tissue necrosis and ulcers [66,67].

Local care after a spider bite comprises rest, ice, compression, and elevation of the affected limb. Drug therapy with dapsone may limit the bite severity and prevent complications. When there is necrosis, the procedures must be surgical and must employ dressings for the healing process [66]. In *Latrodectus mactans* (black widow spider) envenomation, 10% calcium gluconate intravenous, metocarbamol and muscle relaxant are used, due the muscle stiffness that is caused by its venom [68].

In scorpion stings, topical lidocaine and intravenous paracetamol, in addition to the application of ice, is indicated for pain [49].

For local treatment of secondary infections and cellulitis in envenomation, chloramphenicol was not effective in *Bothrops* envenomation, when compared to the control group [44]. The use of prophylactic antibiotics in most snakebites is not supported by available scientific evidence, but some antibiotics, such as penicillin with β-Lactamase inhibitors, clindamycin and metronidazole, can be used in treatments [69]. A clinical trial showed little benefit from the preventive use of amoxicillin with clavulanate in the prevention of secondary infection by *Bothrops* snakes, which corroborates the results of studies by other authors. This scenario suggests that it is still necessary to select antibiotics to be used in future clinical trials for greater evidence of effective treatments for infections as a result of snakebites [63,70].

The only effective treatment for snakebite envenomation is antivenom therapy, but it acts in a systemic rather than local way [71]. In places in which antivenom is unavailable, popular or traditional medicine is often used [14]. A custom transmitted by generations, and widely used by native populations of Africa and Asia and Brazil, is the use of black stone in the treatment of snakebites; however, research demonstrates the ineffectiveness of its application in the patient’s clinical outcome [46,72].

The leaves of *Jatropha molissima* were tested (via intraplantar) in mice exposed to *Bothrops* venoms, and it was observed that when used 30 min earlier, at the concentration of 200 mg/kg, they presented an important inhibitory property and could be used in a complementary manner [36]. *Vellozia flavicans* is a plant with anti-inflammatory properties, and has been tested on *Bothrops* bites. It was found that it was able to neutralize the in vitro neuromuscular blockade of the diaphragmatic muscle of mice, but it does not have antimicrobial activity [37]. In the studies with plants as an alternative treatment, all the authors conclude that there is a need for more scientific research on the subject [13,14,73].

In the literature, it is noted that there are no effective treatments for local manifestations following envenomation; however, in recent years, several studies have maintained their focus on the benefits of phototherapy as a treatment modality to reduce pain, inflammation and edema, promote wound healing in deeper tissues and nerves, and prevent cell death and tissue damage [26,74].

Low-intensity laser treatment has proved itself to be effective in both in vivo and in vitro [75,76]. Studies have suggested a positive effect of photobiomodelating therapy (PBMT) on reducing local pathological effects caused by *Bothrops* venom, and acceleration of myotoxicity-related tissue regeneration [23,32,77]. However, regarding the ideal treatment parameter for clinical application, a protocol for clinical trials in humans has not yet been established [55,78].

Nadur-Andrade et al. (2016) provided a treatment using low-intensity laser in mice, at 30 min and 1 h after the envenomation. The results indicate the reduction of hyperalgesia and inhibition of the nociceptive response in the first and third hour of evaluation [28].

Barbosa (2008) showed that *B. jararacusssu* venom causes significant edema formation between 3 and 24 h after its inoculation, and a compound inflammatory infiltrate, predominantly by neutrophils, was witnessed 24 h after inoculation [22]. The use of low-intensity laser significantly reduced edema formation by 53% and 64%, at 3 and 24 h, respectively, and resulted in a reduction in neutrophil accumulation (*p* < 0.05) [29]. In this sense, laser therapy significantly reduced edema and leukocyte influx in envenomated muscle, which means that this may be a useful future therapy for the local effects caused by *Bothrops* venoms [16,79].

In this line of innovation for possible therapies, hyperbaric oxygen therapy was used effectively in the treatment of snakebites; it has helped to reduce tissue damage, and can often eliminate the need for surgical decompression of an imminent compartmental syndrome. It may act to prevent tissue damage due to inflammation, but it is not a treatment for tissue necrosis [48]. In 2021, an in vivo study examined the effects of ozonized oil therapy (OZT) associated with PBMT on the local effects caused by *B. jararacusssu* venom. The results indicated that individually, PBMT and OZT can partially protect against venom myonecrosis and edema, with OZT being the most effective, especially in early stages after envenomation, corroborating the need for clinical trials in humans to substantiate these new possibilities [80].

One limitation of this review was the non-indication of some of the local treatments adjuvant to antivenom, since most experimental studies are still in preclinical phases. Thus, a meta-analysis becomes untenable as evidence for recommending an intervention to reduce local effects in envenomation. Antivenom, even with a slow local action when compared to the systemic effects, is still the main neutralizer of inflammatory effects and tissue complications.

## 4. Conclusions

This review addresses evidence regarding adjuvant treatments in local damages following envenomation. Although the topic is still under explored, the identifying data found in vitro and in vivo indicate the need for and importance of exploring other local therapies that can complement the use of antivenoms in humans. On the other hand, a few alternative measures are still questionable, such as the use of tourniquets, cryotherapy, antihistamines, and corticosteroids. In respect to natural products (e.g., plants) their clinical findings have not yet been elucidated. Finally, low-intensity laser treatment methods have been demonstrated to be an efficient coadjuvant therapy for envenomation. This systematic review shows that concomitant therapies can contribute positively to controlling the worsening of envenomated tissue, especially in regards to local damage.

## 5. Materials and Methods

This systematic review was performed using the PubMed, MEDLINE, and LILACS databases, and aimed to analyze studies that reported local treatments of wounds and tissue inflammation following envenomation, and mainly used the descriptors “envenomation” and “local treatment” (in English). The period of the search was from 1979 to 2019 (40 years).

The selected articles (n = 39) were read and evaluated by peers, with the inclusion criteria for complete studies being: (a) access to the full content of the article, (b) clinical trial, (c) randomized clinical trial, (d) case report, and (e) prospective study. In summary, the review sought to unite all the procedures that were performed at the site of the envenomation, and aimed to identify the adjuvant treatments used. PRISMA was used as a methodology for the systematic review considering the CHECKLIST, and Figure 2 presents the article eligibility flowchart.

The GRADE [54] scale was used to classify the evidence from the studies as high (High confidence in the correlation between true and estimated effect), moderate (Moderate confidence in the estimated effect, in which it is possible that the true effect is different from the estimated effect), low (Limited confidence in the estimated effect, in which it is possible that the effect may be very different from the estimated effect) or very low (Very little confidence in the estimated effect, in which the effect is very probably different from the estimated effect.). In the pre-clinical studies, the GRADE scale is not applied.

## Figures and Tables

**Figure 1 toxins-15-00313-f001:**
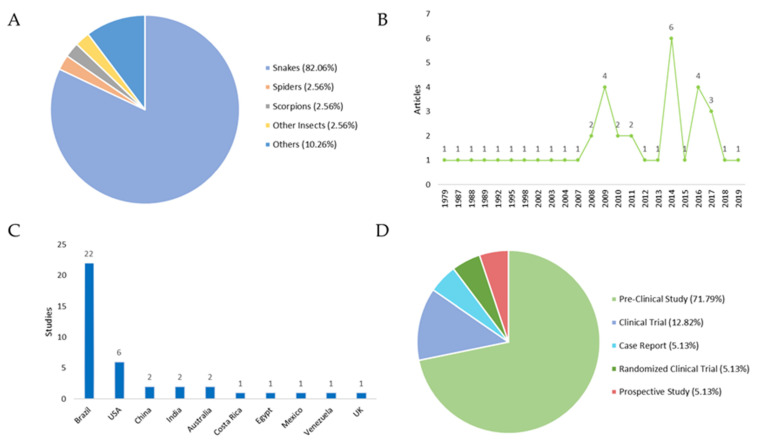
Studies focusing on local therapies following envenomation. (**A**) Distribution of the venomous animals that caused the envenomation. (**B**) Year of publication of the selected studies. Regarding the publication period are addressed in the periods in which these articles were published. An increase in scientific output is observed in the years 2009, 2014, and 2016. (**C**) Number of studies by country. All countries cited in the publications were evaluated by the number of studies, with Brazil and USA producing the most. (**D**) Distribution of study types. In respect to the study type, the following results were obtained: (i) Pre-Clinical Study 71.79%; (ii) Clinical Trials: 12.82%; (iii) Case Reports: 5.13%; Randomized Clinical Trials: 5.13%; and (iv) Prospective Studies: 5.13%.

**Figure 2 toxins-15-00313-f002:**
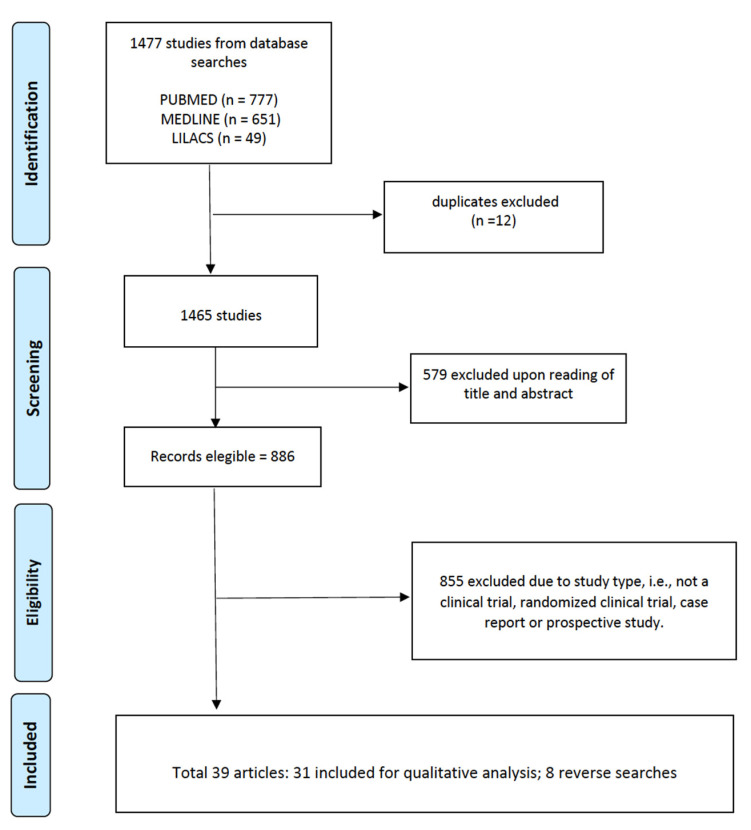
Article eligibility flowchart.

**Table 1 toxins-15-00313-t001:** Pre-clinical studies from 1988 to 2019 focusing on local therapies following envenomation.

Title	Year	Country	Aim	Genus/Species	Ref.
**SNAKES**					
Effects of constriction bands on rattlesnake venom absorption: a pharmacokinetic study	1992	USA	Determine whether the use of a constriction band alters systemic absorption of rattlesnake venom in pigs and whether the use of constriction bands alters local swelling	*Crotalus atrox*	[18]
Rationalisation of first-aid measures for elapid snakebite	1979	Australia	Investigate the treatment for snakebite in monkeys (the anatomical similarities of their limbs to those of man allows the application of relevant first-aid measures to the envenomated limb) by radioimmunoassay monitoring of the distribution of whole-venom components and a neurotoxin in envenomated animals	*Crotalus adamanteus*	[19]
Electric shock does not save snake bitten rats	1988	USA	Verify the application of a series of high voltage, low magnification, direct current shocks directly to a snakebite site on a rat	*Bothrops atrox*	[20]
Analgesic effect of Light-Emitting Diode (LED) therapy at wavelengths of 635 and 945 nm on *Bothrops moojeni* venom-induced hyperalgesia	2014	Brazil	Evaluate the effectiveness of the light-emitting diode with 635 and 945 nm wavelengths in reducing inflammatory processes and hyperalgesia induced by *Bothrops moojeni* venom in mice	*Bothrops moojeni*	[21]
Effect of low-level laser therapy in the inflammatory response induced by *Bothrops jararacussu* snake venom	2008	Brazil	Report the effect of low power laser therapy on the formation of edema and leukocyte absorption caused by *Bothrops jararacussu* venom as an alternative treatment for *Bothrops* snakebites.	*Bothrops jararacussu*	[22]
Effects of a low-level semiconductor gallium arsenide laser on local pathological alterations induced by *Bothrops moojeni* snake venom	2013	Brazil	Investigate the effect of a low-power semiconductor gallium arsenide laser on local pathological changes induced by *B. moojeni* venom	*Bothrops moojeni*	[23]
Effects of photobiostimulation on edema and hemorrhage induced by *Bothrops moojeni* venom	2012	Brazil	Study the effectiveness of low-power laser and light-emitting diode irradiation alone or in combination with AV in the local reduction of edema and hemorrhage induced by *Bothrops moojeni* venom in mice	*Bothrops moojeni*	[24]
Low-intensity laser therapy improves tetanic contractions in mouse anterior tibialis muscle injected with *Bothrops jararaca* snake venom	2016	Brazil	Examine the influence of low-intensity laser therapy on the contractile activity of the mouse muscle injected with *Bothrops jararaca* venom	*Bothrops jararaca*	[25]
Low-level laser therapy decreases local effects induced by myotoxins isolated from *Bothrops jararacussu* snake venom	2010	Brazil	Analyze the effect of low-level laser therapy on the formation of edema, leukocyte influx, and myonecrosis caused by BthTX-I and BthTX-II, isolated from *Bothrops jararacussu* venom	*Bothrops jararacussu*	[16]
Low-level laser therapy reduces edema, leukocyte influx and hyperalgesia induced by *Bothrops jararacussu* snake venom	2011	Brazil	Investigate the effect of low-level laser therapy on inflammatory levels and hyperalgesia induced by *B. jararacussu* venom	*Bothrops jararacussu*	[26]
The ability of low level laser therapy to prevent muscle tissue damage induced by snake venom	2009	Brazil	Analyze the ation in the nerve-muscle by *Bothrops jararacussu* venom	*Bothrops jararacussu*	[27]
Analgesic effect of photobiomodulationon *Bothrops moojeni* venom induced hyperalgesia: a mechanism dependent on neuronal inhibition, cytokines and kinin receptors modulation	2016	Brazil	Evaluate the efficacy of photobiomodulation to reduce inflammatory hyperalgesia induced by *Bothrops moojeni* venom	*Bothrops moojeni*	[28]
Effect of low-level laser therapy in the myonecrosis induced by *Bothrops jararacussu* snake venom	2009	Brazil	Evaluate the ability of low-level laser therapy, alone or in combination with antivenom, to reduce venom-induced myonecrosis in rats	*Bothrops jararacussu*	[29]
Effects of photobiomodulation therapy on *Bothrops moojeni* snake-envenomed gastrocnemius of mice using enzymatic biomarkers	2017	Brazil	Evaluate the daily irradiation with neon helium laser and semiconductor laser of gallium arsenide at the site of intramuscular injection of venom in the gastrocnemius muscle of rats	*Bothrops moojeni*	[30]
Effects of the Ga-As laser irradiation on myonecrosis caused by *Bothrops moojeni* snake venom	2003	Brazil	Evaluate the effects of the gallium arsenide laser on gastrocnemius muscle in envenomated mice	*Bothrops moojeni*	[17]
Low-level laser therapy (904 nm) counteracts motor deficit of mice hind limb following skeletal muscle injury caused by snakebite-mimicking intramuscular venom injection	2016	Costa Rica	Investigate the motor function behavior of mice subjected to a *Bothrops jararacussu* snake venom injection and exposed to low-power laser therapy	*Bothrops jararacussu*	[31]
Low-level laser therapy promotes vascular endothelial growth factor receptor-1 expression in endothelial and nonendothelial cells of mice gastrocnemius exposed to snake venom	2011	Brazil	Evaluate the ability of low-power laser therapy to promote angiogenesis and myoregeneration in mice	*Crotalinae* and *Bothrops moojeni*	[32]
Photobiostimulation reduces edema formation induced in mice by Lys-49 phospholipases A_2_ isolated from *Bothrops moojeni* venom	2014	Brazil	Analyze the effect of a low-power laser (LED) after edema formation caused by snake venom in rats	*Bothrops moojeni*	[33]
The effects of low-level laser on muscle damage caused by *Bothrops neuwiedi* venom	2008	Brazil	Analyze the effects of low-power laser on myonecrosis induced by *Bothrops neuwiedi* venom in rats	*Bothrops neuwiedi*	[34]
Experimental *Bothrops atrox* envenomation: efficacy of antivenom therapy and the combination of *Bothrops* antivenom with dexamethasone	2017	Brazil	Test the effectiveness of *Bothrops* antivenom in treating signs, symptoms and toxic effects induced by *B. atrox* venom in mice	*Bothrops atrox*	[5]
Rosemary leaves extract: anti-snake action against egyptian *Cerastes cerastes* venom	2018	Egypt	Explore the neutralization capacity of the aqueous extract of leaves of *Rosmarinus officinalis* L. (RMAE) against Egyptian *Cerastes cerastes* (Cc) viper venom	*Cerastes cerastes*	[35]
Aqueous leaf extract of *Jatropha mollissima* (Pohl) bail decreases local effects induced by bothropic venom	2016	Brazil	Evaluate the effect of the aqueous extract of leaves of *J. mollissima* on the local effects induced by *Bothrops* venoms	*Bothrops erythromelas* and *Bothrops jararaca*	[36]
*Vellozia flavicans* Mart. Ex Schult. Hydroalcoholic extract inhibits the neuromuscular blockade induced by *Bothrops jararacussu* venom	2014	Brazil	Observe possible antimicrobial activities of *V. flavicans* against snake venom	*Bothrops jararacussu*	[37]
Antitoxin activity of aqueous extract of *Cyclea peltata* root against *Naja naja* venom	2017	India	Test the neutralization potential of *Cyclea peltata* root extract via ex vivo and in vivo approaches for *Naja naja* venom	*Naja naja*	[38]
*Abarema cochliacarpos* extract decreases the inflammatory process and skeletal muscle injury induced by *Bothrops leucurus* venom	2014	Brazil	Evaluate the anti-ophidic capacity of *A. cochliacarpos* extract and compare the activity of the extract with dexamethasone	*Bothrops leucurus*	[39]
Small incisions combined with negative-pressure wound therapy for treatment of *Protobothrops Mucrosquamatus* bite envenomation: a new treatment strategy	2019	China	Evaluate the effects of a new treatment strategy for envenomation, which consists of multiple small incisions and negative pressure wound therapy (NPWT) on the swelling of the injured limbs and the systemic inflammatory reaction	*Protobothrops mucrosquamatus*	[40]
Inhibitory effect of plant *Manilkara subsericea* against biological activities of *Lachesis muta* snake venom	2014	Brazil	Investigate in vitro and in vivo the ability of different ethanolic extracts of leaves and stems of *M. subsericea* and solvent-partitioned fractions to neutralize some biological activities induced by the *L. muta* venom	*Lachesis muta*	[41]
**SPIDERS**					
Therapy of brown spider envenomation: a controlled trial of hyperbaric oxygen, dapsone, and cyproheptadine	1995	USA	Determine whether hyperbaric oxygen, dapsone, or cyproheptadine decreases the severity of skin lesions resulting from an experimental *Loxosceles* envenomation	*Laxosceles*	[42]

**Table 2 toxins-15-00313-t002:** Studies from 1988 to 2019 focusing on local therapies following envenomation.

Title	Year	Type of Study	Country	Aim	Animal	Ref.	GRADE Scale *
**INSECTS**							
Venom immunotherapy reduces large local reactions to insect stings	2009	Clinical trial	USA	Determine the feasibility of performing a controlled trial to examine the efficacy of venom immunotherapy in reducing the size and duration of large local reactions	*Hymenoptera*	[43]	High
**SNAKES**							
Failure of chloramphenicol prophylaxis to reduce the frequency of abscess formation as a complication of envenoming by *Bothrops* snakes in Brazil: a double-blind randomized controlled trial	2004	Randomize double-blind clinical trial	Brazil	Assess whether antibiotic therapy is effective in preventing infection as a complication of envenomation	*Bothrops*	[44]	High
The efficacy of tourniquets as a first-aid measure for Russell’s viper bites in Burma	1987	Clinical trial	UK	Evaluate the effectiveness of the tourniquet, which is commonly used as a first-aid measure in victims of Russell’s viper bites	*Vipera russelli siamensis*	[45]	Moderate
Study of the efficacy of the black stone on envenomation by snake bite in the murine model	2007	Case report	Mexico	Evaluate the effectiveness of black stone against snake venoms	*Bitis arietans, Echis ocellatus* and *Naja nigricollis*	[46]	Very low
Tourniquet ineffectiveness to reduce the severity of envenoming after *Crotalus durissus* snake bite in Belo Horizonte, Minas Gerais, Brazil	1998	Case report	Brazil	Describe whether tourniquet use reduces the severity of envenomations following a *Crotalus durissus* snakebite	*Crotalus durissus*	[47]	Very low
A multidisciplinary approach with hyperbaric oxygen therapy improves outcome in snake bite injuries	2015	Prospective study	India	Treatment of snakebite injuries in the extremities with various treatment modalities, including hyperbaric oxygen (HBO) therapy, surgical debridement, skin grafts, local or distant flaps to provide effective treatment from the perspective of plastic surgeons	Not specified	[48]	Low
**SCORPIONS**							
A randomized trial comparing intravenous paracetamol, topical lidocaine, and ice application for treatment of pain associated with scorpion stings	2014	Clinical trial	USA	Compare the effectiveness of three treatment modalities in patients with pain caused by scorpion bites using visual analog scale (VAS) scores	*Androctonus crassicauda*	[49]	High
**OTHERS**							
Cold packs: Effective tropical analgesia in the treatment of painful stings by *Physalia* and other jellyfish	1989	Clinical trial	Australia	Test whether application of cold packs is effective as topical analgesia in relieving mild to moderate pain in jellyfish stings	*Physalia*	[50]	High
A randomized paired comparison trial of cutaneous treatments for acute jellyfish (*Carybdea alata*) stings	2002	Randomized study	USA	Compare cutaneous treatments (heat, papain, and vinegar) for acute jellyfish (*Carybdea alata*) stings	*Carybdea alata*	[51]	Moderate
Comparisons of ice packs, hot water immersion, and analgesia injection for the treatment of centipede envenomations in Taiwan	2009	Clinical trial	China	Compare the effectiveness of ice packs and hot water immersion in treating centipede envenomation	*Centipede*	[52]	High
Clinical, epidemiological and treatment aspects of five cases of sea urchin poisoning in Adícora, Paraguaná Peninsula, Falcón State, Venezuela	2010	Descriptive and prospective study	Venezuela	Analyze clinical aspects, epidemiology and treatment of envenomation caused by sea urchins in Venezuela	*Lytechinus variegatus* and *Echinometra lucunter*	[53]	Low

* The GRADE [54] scale was used to classify the evidence from the studies as high (High confidence in the correlation between true and estimated effect), moderate (Moderate confidence in the estimated effect, in which it is possible that the true effect is different from the estimated effect), low (Limited confidence in the estimated effect, which may be very different from the estimated effect) or very low (Very little confidence in the estimated effect, which is very probably different from the estimated effect).

**Table 3 toxins-15-00313-t003:** Local treatments according to the animal and indicated and contraindicated or ineffective treatments.

Animal	Treatment	
	Indicated	Contraindicated or Ineffective
Snakes	Aqueous extract of rosemary in vivo [35], *Jatropha mollissima* Pohl leaf aqueous extract in vivo [36], *Vellozia flavicans* hydroalcoholic extract in vitro [37], root extracts of *C. peltate* [38], hydroethanol extract of *Abarema cochliacarpos* [39], *Andrographis peniculata* and *Polygonum cuspidatum* extract [41], low-level laser (reduction in inflammatory signs in vivo and myonecrosis in vitro and in vivo) [16,21,22,23,25,26,27,28,29,30,31,32,33,34,55], intraperitoneal dexamethasone in vivo [5], hyperbaric oxygen [48], and incisions combined with negative pressure in wound therapy [40]	Chloramphenicol is not effective for secundary infection [44], tourniquet is contraindicated for envenomation [47,56], and black stone application is not effective for envenomation [46]
Scorpion	Topical lidocaine and ice [49]	-
Others (jellyfish, centipedes and sea urchins)	Ice pack for centipede envenomations [52], and topical application of iodinated antiseptic solution, local anesthetic, analgesic, anti-inflammatory and antibiotic therapy for sea urchins [53] and jellyfish [50]	-

**Table 4 toxins-15-00313-t004:** Outcomes of the local treatments used in the evaluated studies.

Treatment	Conclusion	Ref.
Application of venom immunotherapy in reducing the size and duration of large local reactions to insect stings.	The therapy significantly reduced the size and duration of the large local reactions, and the efficacy improved over a period of 2 to 4 years of treatment.	[43]
Venom of the fer-de-lance, *Bothrops atrox*, was injected subcutaneously into rats in a series of increasing doses.	Envenomated animals developed hemorrhagic ulcers at the injection sites, the size of which was strongly related to venom dose.	[20]
To assess the severity of skin lesions resulting from *Loxosceles* envenomation, New Zealand white rabbits were used. All groups received 20 micrograms of pooled *L. deserta* venom intradermally. The control group received 4 mL of a 5% ethanol solution by oral gavage every 12 h for 4 days. The HBO group received hyperbaric oxygen at 2.5 ATA for 65 min every 12 h for 2 days, plus 5% ethanol solution for 4 days. The dapsone group received dapsone 1.1 mg/kg in 5% ethanol by gavage every 12 h for 4 days. The cyproheptadine group received cyproheptadine. 125 mg/kg in 5% ethanol by gavage every 12 h for 4 days.	Total lesion size and ulcer size were followed for 10 days. The lesions were then excised, examined microscopically, and ranked by the severity of the histopathology. The groups did not differ significantly with respect to lesion size, ulcer size, or histopathologic ranking. Given the negative result in this study with adequate power to detect meaningful treatment benefits, hyperbaric oxygen, dapsone, or cyproheptadine are not recommended in the treatment of *Loxosceles* envenomations.	[42]
Using a crossover design, five pigs were studied with and without the use of a constriction band. First, 125l-Labeled Western Diamondback rattlesnake (*Crotalus atrox*) venom was injected subcutaneously into one foreleg. The protocol was repeated using the opposite foreleg six days later. The constriction band and leg circumference were measured serially.	The use of a constriction band was effective in reducing venom absorption while it was in place (reduced area under the venom concentration-time curve and maximum plasma venom concentration in the cuffed group), and constriction band removal did not result in a significant increase in maximum plasma venom concentration. Leg swelling was not affected by constriction band use. Because constriction band use delayed venom absorption without causing increased swelling, it may prove to be a useful first aid measure in human beings.	[18]
Monkeys envenomated with tiger snake (*Notechis scutatus*) venom were monitored by radioimmunoassay for both crude venom and a neurotoxin.	Venom movement can be effectively delayed for long periods by the application of a firm crepe bandage to the length of the bitten limb combined with immobilization by a splint. Pressure alone or immobilization alone did not delay venom movement.	[19]
Oral antibiotic prophylaxis with chloramphenicol in the prevention of infection in *Bothrops* snakebite envenomation.	Chloramphenicol for *Bothrops* snakebite victims with signs of local envenomation is not effective for preventing local infections.	[44]
Application of black stone after injection of intramuscular venom of *Bitis arietans, Echis ocellatus and Naja nigricollis* in rats.	Showed no effects on envenoming outcome.	[46]
Tourniquet use after *Crotalus* envenomations.	The data demonstrated the ineffectiveness of the use of the tourniquet to reduce the severity of envenomations by *Crotalus.*	[47]
Evaluation of the efficacy of light-emitting diode (LED) use in reducing inflammatory hyperalgesia induced by *Bothrops moojeni* venom in mice.	LED therapy is effective in reducing venom-induced hyperalgesia even after symptoms are present.	[21]
Evaluation of the effect of low-level laser therapy (LLLT) on edema formation and leukocyte influx caused by *Bothrops jararacussu* venom as an alternative treatment for *Bothrops* snakebites.	LLLT reduced edema and leukocyte influx, which suggests that LLLT should be considered as a potential therapy for the local effects of *Bothrops* envenomations.	[22]
Investigation of the effect of a low-intensity semiconductor gallium arsenide (GaAs) laser on local pathological alterations induced by *B. moojeni* venom.	Laser irradiation can help to reduce some local effects as it stimulates muscle fiber regeneration.	[23]
Evaluation of the effectiveness of LLLT and LED with and without antivenom for reducing local edema formation and hemorrhage induced by *Bothrops moojeni* venom in mice.	Laser and LED irradiation reduced venom-induced local effects in combination with antivenom.	[24]
The influence of LLLT on the contractile activity of the skeletal muscle of mice injected with *Bothrops jararaca* venom was examined, in laser with antivenom and only laser groups.	Antivenom and LLLT treatment groups improved muscle function after venom-induced damage.	[25]
Evaluation of the effect of LLLT, at a dose of 4.2 J/cm^2^, on edema formation, leukocyte influx and myonecrosis caused by *Bothrops jararacussu* venom.	LLLT significantly reduced edema formation, neutrophil accumulation, and myonecrosis induced by both myotoxins 24 h after envenomation.	[16]
Use of LLLT to reduce inflammatory signs in snakebite envenomations.	The group of mice treated with LLLT had a significant reduction in inflammatory signs.	[26]
Evaluation of the effects of neon helium laser (HeNe) at three energy densities on *Bothrops jararacussu* envenomations in rats.	HeNe laser irradiation, at a dosage of 3.5 J/ cm^2^, effectively reduces myonecrosis, and the blocking effect of neuromuscular transmission.	[27]
The efficacy of laser therapy, with energy of 2.2 J/cm^2^, to reduce hyperalgesia induced by *Bothrops moojeni* venom in mice was evaluated.	The use of photobiomodulation in the reduction of local pain induced by *Bothrops* venom was considered effective.	[28]
Investigation of the ability of LLLT alone or in combination with antivenom to reduce myonecrosis induced by *Bothrops jararacussu* venom.	Antivenom therapy alone was ineffective in reducing myonecrosis, but with LLLT it significantly reduced myonecrosis of the envenomated muscle.	[29]
The effectiveness of HeNe and GaAs lasers on local effects in rats injected with *Bothrops moojeni* venom was evaluated.	Histopathological analysis revealed increased muscle regeneration in groups of rats treated with both lasers; however, GaAs laser showed the best results.	[30]
The effects of GaAs lasers were evaluated in mice that had *Bothrops moojeni* venom injected intramuscularly.	GaAs irradiation significantly decreased the amount of myonecrosis in all tests performed.	[17]
Motor function was analyzed in mice submitted to injection of *Bothrops jararacussu* venom and exposed to LLLT at 3, 24, 48 and 72 h after envenomation	The motor function of the mice improved after the first laser application.	[31]
Evaluation of whether LLLT could accelerate angiogenesis and myoregeneration in mice injected with *Bothrops moojeni* venom.	In 3 days, LLLT increased angiogenesis, decreased neutrophils and increased proliferation of regenerating cells, i.e., improved revascularization.	[32]
The effect of LLLT and LED on edema formation in envenomated mice was analyzed.	Both LLLT and LED were similar in reducing edema formation, the effect being greater when photobiostimulation was combined with antivenom.	[33]
Evaluation of the effects of a low-intensity laser on myonecrosis caused by the insertion of *Bothrops neuwiedi* venom in the gastrocnemius muscle of rats.	Low-intensity laser reduced neutrophilic inflammation, as well as myofibrillar edema, hemorrhage, and myonecrosis, which suggests that laser therapy may be useful as an adjunctive therapy after *Bothrops* envenomations.	[34]
Antivenom administered intravenously and dexamethasone administered intraperitoneally were tested to analyze edema and muscle damage caused by *Bothrops atrox* envenomation.	The use of dexamethasone, associated with antivenom, reduced recovery time after edema and muscle injury.	[5]
Hyperbaric oxygen therapy (HBO) was performed during 90 min for 6 days in patients with cellulite and compartment syndrome. Patients with soft tissue necrosis underwent surgical debridement and soft tissue reconstruction.	Adjunctive HBO therapy has been found to be effective in treating snakebite injuries. The authors suggest further research.	[48]
To verify the venom inhibition using 30 µg of raw Egyptian viper venom in mice, different proportions of aqueous extract of rosemary leaves were injected for 30 min at 37 °C.	Rosemary leaf extract has a potential neutralizing action against local effects and lethality of *Cerastes cerastes* venom. Those who received the extract showed an increase in survival time when compared to the control.	[35]
Aqueous extract of *Jatropha molissima* Pohl leaves was injected in different doses in a group of five mice and, after 30 min, the animals received a subcutaneous injection of 25 µg of *B. erythromelas* and *B. jararaca* venoms.	*J. mollissima* has substances that can inhibit the toxins present in the venoms. However, further experiments are needed to define its adjuvant action in the treatment of local effects.	[36]
The capacity of the hydroalcoholic extract of *Vellozia flavicans* to neutralize the neuromuscular blockade in vitro caused by the venom of *Bothrops jararacussu* in mice and its antimicrobial capacity was also tested.	The hydroalcoholic extract of *V. flavicans* leaves was effective in neutralizing and decreasing in vitro the neuromuscular blockade caused by *B. jararacussu*. However, it has no significant antimicrobial activity against the tested bacteria.	[37]
The venom toxicity was evaluated in neutralization assays. Root extracts of *Cyclea peltata* were used to evaluate neutralization in ex vivo and in vivo tests using assays to determine acetylcholinesterase, protease, direct hemolysis, phospholipase activity and procoagulant activity.	The result of the ex vivo and in vivo analysis indicates that the root extract of *C. peltata* has significant compounds that can neutralize toxins from *N. naja* venom.	[38]
The treatment consisted of the application of hydroethanolic extract of *Abarema cochliacarpos* in mice by oral probe to evaluate the effects of *Bothrops leucurus* envenomation.	The crude extract of *A. cochliacarpos* was able to reduce the edematogenic effect and the hyperalgesic action of the venom.	[39]
Small incisions combined with negative pressure wound therapy (NPWT) for treatment of *Protobothrops mucrosquamatus* envenomations as a new treatment strategy.	Multiple small incisions combined with NPWT have been shown to be effective in controlling the release of inflammatory cytokines and accelerating the relief of the inflammatory reaction.	[40]
Via a survey of the literature, the authors found two species of plants (*Andrographis peniculata* and *Polygonum cuspidatum*) used topically in the healing of wounds after snakebite.	Upon review, the extracts proved to be effective as possible adjuvant treatments in the healing of snakebite wounds.	[41]
A randomized study carried out over a period of 1 year in patients with scorpion stings, who did not have systemic signs or symptoms. Patients were treated with intravenous paracetamol, topical lidocaine and the application of ice.	Topical lidocaine and ice were considered to be effective and safe treatments for scorpion stings with regard to pain in patients without systemic signs and symptoms.	[49]
Comparison of the effectiveness of ice packs and hot water immersion for the treatment of pain after centipede envenomation. Sixty centipede-envenomated patients were randomized into three groups and treated with ice packs, hot water immersion or analgesic injection.	Ice packs and soaking in hot water can reduce pain in centipede envenomations. Their effects seem to be equivalent to analgesics, but the study suggests that ice has the best cost effectiveness.	[52]
Treatment consisted of topical application of an iodinated antiseptic solution, local anesthetic, analgesic, intravenous anti-inflammatory, antibiotic therapy and tetanus therapy after sea urchin envenomation.	The patients evolved satisfactorily between 20 and 45 min after the beginning of the therapeutic treatment, and presented a low pain score. The wounds healed without edema, erythema, necrosis or bacterial growth.	[53]
Venom antigen levels were measured in blood samples obtained above and below a tourniquet, and before and after their release in human victims of Russell’s viper (*Vipera russelli swnensis*) envenomation, to discover whether the tourniquet retarded the proximal spread of venom from the site of the bite.	The efficacy of the tourniquets, which are commonly used, was studied in 37 cases by measuring venom antigen levels. In most cases, the tourniquet did not prevent proximal spread of venom. In 8/37 cases, however, venom antigen assays suggested but did not prove that venom absorption was being delayed by the tourniquet.	[45]
Application of cold packs as a topical analgesia is effective in relieving mild-to-moderate pain in jellyfish stings.	This study has shown that, when applied to *Physalia* (“bluebottle”) jellyfish stings, cold packs are effective as topical analgesia in the relief of mild-to-moderate skin pain. The application of ice also has been shown to be effective for topical analgesia in a number of other jellyfish stings, including those by *Cyanea* (“hair jellyfish”), *Tamoya sp.* (“Moreton Bay stinger” or “fire jelly”) and *Carybdea rastoni* (“jimble”) as well as by *Physalia.*	[50]
Treatment consisted of cutaneous treatments (heat, papain and vinegar) for acute jellyfish (*Carybdea alata*) stings.	This study suggests that the most efficacious initial treatment for *C. alata* jellyfish envenomation is hot-water immersion of the afflicted site.	[51]

## Data Availability

Data sharing not applicable.

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
