# Peer review of "Prospecting Local Treatments Used in Conjunction with Antivenom Administration Following Envenomation Caused by Animals: A Systematic Review"

_toxins, 2023, doi:10.3390/toxins15050313_

Round 1
Reviewer 1 Report
This is a generally well structured and executed review article, and it is enjoyable to read in its timely and comprehensive scope.
Major point:
In the case of envenomation, "local" can have two implications. First, the "first-aid" treatment(s) applied at the geographic site of the emergency event, most often administered by healthcare amateurs, and second, the treatment applied at hospital clinic by professionals. I feel this paper would increase its impact in the literature by distinguishing these two "local" roles. I suggest a call-out table or figure in Conclusions where we could see science-supported actions in "first-aid in the field", followed by science supported fast treatments administered in hospital clinic. Such a piece will help the reader, but will also help the field conceptualize current observations that appear to contradict each other, but possibly do not. One example is the controversial topic of first-aid techniques to apply "direct pressure" to inhibit blood flow from the site of envenomation. This paper cites controlled pre-clinical data showing a positive effect in case of pigs and monkeys using constriction bands or crepe bandages. Yet two citations (Refs. 47, 55) regarding tourniquet usage in humans sees lack of efficacy, or worse, prolonged application and gangrene risk. But the human data comes from amateur-applied tourniquets, whereas the animal data comes from professionally applied techniques. Thus there are at least two variables that could explain discrepant results: either constriction bands and crepe bandages are actually more effective than tourniquets, or alternatively all of these could work well but not when amateurs, non-professionals, apply them, on average. How could this review article handle the issue? One way is to directly state what should be in the first-aid kit: perhaps constriction bands and crepe bandages (not tourniquets), perhaps botanical agents specific to venoms typical of a specific region, etc. Items and practices for which this review documents positive scientific data. Can low-intensity laser by administered in the field? If so, let this be stated and included in a field kit. Next, local hospital clinic would have its specific professionally applied options: anti-venom antisera, laser options, surgery, etc.
Minor comments:
(2) Defining "grade scale" should be included in figure/table legend. The definition is at the end of the paper, currently.
(3) Figure 1, the color scales in pie charts should have greater difference in the color scheme. I cannot distinguish the three most intense blues or greens from each other.
(4) "contraindication" should be distinguished from "not effective", where the former means the action or agent increases risk of poor outcomes, while the latter means there is neither positive nor negative effect. Table 3, for example.
(5) Suggested edit for line 68, "Therefore, local adjuvant treatments are needed to mitigate the local damage following venomous animals’ aggression and envenoming", to something similar to "... identification of best practices for local adjuvant treatments could help to mitigate local damage following venomous animals' aggression and envenoming".
Author Response
We appreciate all considerations made by the reviewer for the manuscript and hope to meet the journal's requirements for publication.
In this systematic review, we explore therapies that address the local effects caused by venoms from venomous animals. Based on studies and procedures performed on local injury after envenomation, we discuss adjuvant therapeutic techniques for the treatment of local lesions to be developed by health professionals. The intent of this article is to help inform professional clinical practice guidelines and improve patient outcomes in regions where venomous animals are prevalent. Thus, the antivenoms and procedures listed in this article are not indicated for care as first aid in a pre-hospital environment, nor to be used as self-administered procedures by patients.
Minor comments:
(2) Defining "grade scale" should be included in figure/table legend. The definition is at the end of the paper, currently.
Response: We appreciate the recommendation. In Table 2, it was indicated with * the reference and classification of the scale GRADE. The Figure 1 was not considered this scale of evidence for the construction of graphs.
(3) Figure 1, the color scales in pie charts should have greater difference in the color scheme. I cannot distinguish the three most intense blues or greens from each other.
Response: We appreciate the recommendation and We've adjusted the colors in Figure 1 as requested.
(4) "contraindication" should be distinguished from "not effective", where the former means the action or agent increases risk of poor outcomes, while the latter means there is neither positive nor negative effect. Table 3, for example.
Response: We appreciate the recommendation and We adjust table 3 as requested.
(5) Suggested edit for line 68, "Therefore, local adjuvant treatments are needed to mitigate the local damage following venomous animals’ aggression and envenoming", to something similar to "... identification of best practices for local adjuvant treatments could help to mitigate local damage following venomous animals' aggression and envenoming".
Response: We appreciate the recommendation and the text has been adjusted accordingly.

Reviewer 2 Report
Envenomings caused by venomous animals are a significant global public health concern, particularly in regions where venomous animals are prevalent. Antivenom administration is the primary treatment for envenomings caused by venomous animals, and it has been shown to be effective in reducing morbidity and mortality associated with envenomings. However, there is growing interest in prospecting local treatments in addition to antivenom administration for envenomings caused by venomous animals. This is because in some cases, antivenom may not be available or may not provide complete protection against the venom. Additionally, some local treatments may have a complementary effect and may reduce the severity of envenomings. There are various local treatments that have been proposed for envenomings caused by venomous animals, including the use of traditional medicinal plants, cold compresses, tourniquets, and pressure immobilization. However, the efficacy and safety of these local treatments are not well established, and some may even be harmful if used improperly.
In this systematic review, the authors explored therapies targeting local effects caused by venomous animal envenomings. Based on studies and procedures performed on local injuries following envenomings, the authors have discussed adjuvant therapeutic techniques for the treatment of local injuries. The systematic review provides a comprehensive and unbiased summary of the available evidence on the efficacy and safety of local treatments in addition to antivenom administration for envenomings caused by venomous animals. This could help inform clinical practice guidelines and improve patient outcomes in regions where venomous animals are prevalent.
Overall, the review is well organized and presented and I recommend its acceptance for publication after addressing a few concerns as commented below.
1. The following species name should be italicized; Jatropha molissima in line 182, Solenopsis invicta in line 142, Crotalus durissus in line 120, Vellozia flavicans in line 185, Physalia physalise in line 129, Loxosceles inmates in line 155 and B. jararacusssu in line 217. Generally, ensure all species names in the entire document are scientifically written.
2. Please provide the citations for the following sentences in,
i. Lines 46-47
ii. Lines 126-127
iii. Lines 135-136
3. Pay attention to the spacing between words such as line 43
4. ‘in vivo’ and ‘in vitro’ should be written in a standard format and uniformly in the entire document
5. Are there any particular venomous animals for which local treatments in addition to antivenom have been found to be especially ineffective? Please mention and discuss any relevant literature in this manuscript, if any.
6. If there are any findings of potential risks or side effects of using local treatments in addition to antivenom therapy, it is recommended that the author mention and discuss them.
7. Even though the standard of scientific writing is generally good, some of the sentences and language use are unclear. English grammar polish would be beneficial to the overall manuscript.
Author Response
We greatly appreciate the manuscript considerations and being given the opportunity to meet the requirements for publication in the journal.
- The following species name should be italicized; Jatropha molissima in line 182, Solenopsis invicta in line 142, Crotalus durissus in line 120, Vellozia flavicans in line 185, Physalia physalise in line 129, Loxosceles inmates in line 155 and B. jararacusssu in line 217. Generally, ensure all species names in the entire document are scientifically written.
Response: We thank you and adjust according to the recommendation
- Please provide the citations for the following sentences in,
- Lines 46-47
- Lines 126-127
- Lines 135-136
Response: We thank you and adjust according to the recommendation
- Pay attention to the spacing between words such as line 43
Response: We thank you and adjust according to the recommendation
- ‘in vivo’ and ‘in vitro’ should be written in a standard format and uniformly in the entire document
Response:
- Are there any particular venomous animals for which local treatments in addition to antivenom have been found to be especially ineffective? Please mention and discuss any relevant literature in this manuscript, if any.
Response: We appreciate this observation and have adjusted the term contraindication to not effective in Table 3 to understand the terminologies better and thus be more consistent with the discussion.
- If there are any findings of potential risks or side effects of using local treatments in addition to antivenom therapy, it is recommended that the author mention and discuss them.
Response: We appreciate the observation and verified that the authors of the articles did not mention the risks of the use of therapies in preclinical studies or in clinical studies.
- Even though the standard of scientific writing is generally good, some of the sentences and language use are unclear. English grammar polish would be beneficial to the overall manuscript.
Response: We will forward the final version for further English language review.

Round 2
Reviewer 1 Report
The choices and edits made by the authors to clarify this interesting article are adequate for acceptance.